# Thornton Wilder and Arthur Miller: A Brief History of Time, Space, and Matter

**Salvatore Talluto** 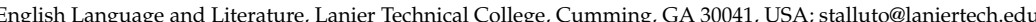

English Language and Literature, Lanier Technical College, Cumming, GA 30041, USA; stalluto@laniertech.edu

**Abstract:** While many literary theories focus on materialistic concerns, less frequently have these theories focused on the spiritual matters arising from such concerns. A cosmological interpretive strategy focuses on such spiritual and cosmic themes rather than ignoring them. This essay's analysis will focus on using a cosmological interpretive strategy to analyze Thornton Wilder's *Our Town* and Arthur Miller's *The Man Who Had All the Luck*. This strategy will reveal that, rather than merely being focused on spatial and material concerns, these texts also demonstrate a concern with our relationships with nature and the wider cosmos. Through their narratives, both Wilder and Miller address the passage of time and the questions of agency which occur when thinking about time. This analysis will demonstrate how these stories deny economic and historical determinism in favor of an interdependency between humans and the wider cosmos. These texts help reveal reality as a set of interconnected narratives and histories that include each individual, the societies around that individual, nature around those societies, and the wider cosmos within which everything exists.

**Keywords:** Thornton Wilder; Arthur Miller; spirituality; cosmology; cosmological; disenchantment; re-enchantment; modernism; postmodernism



## 1. Introduction

Thornton Wilder's *Our Town* and Arthur Miller's *The Man Who Had All the Luck* are not often thought of together, or seen as both being deeply spiritual. A cosmological interpretive strategy analyzes artistic creation as an attempt to possibly mimic, in micro, the cosmic creation in order to understand the place of the individual within human society and the wider cosmos. This strategy focuses on analyzing conceptions of time, conceptions about relationships to nonhuman entities, and conceptions about agency and fate. A cosmological interpretive framework analyzes how the relationships between human constructions of time and nonhuman constructions of time, including geological time, to arrive at a conception of time from a cosmic standpoint. In addition, this strategy looks at ways in which otherness can be overcome to recognize an interdependence between human life and cosmic creation. Awareness of cosmic time and the interdependence of human life help to lead readers to see how our agency and fate is not controlled through deterministic factors but is created in reaction to, through, and in concert with the interdependent cosmos that we are a part of. This creation of reality through a comingling of human and cosmic forces working interdependently together, rather than in opposition to each other, is called theopoetics. These themes and the theopoetical outcomes within these two works of literature are born out of the struggle of the characters with feelings of disconnect and dislocation produced by factors such as factory work, technological advancements such as the automobile, and the traumatic experiences of World War I and World War II. The effects of such disconnect and dislocation in these texts mirror Max Weber's descriptions of the process of rationalization in his essay "Religious Rejections of the World and Their Directions." Weber labels rationalization as the process through which logic and scientific thinking became separated from spiritual and cosmological thinking. This essay has become a foundation for theorists to critique so-called modernity and post-modernity as leading to

a destructive commodification of nature, culture, and even life itself. Weber argues that rationalization is the basis for this destructive commodification. He also argues that many people have become alienated from society and their own agency due to this rationalization, a process that Weber calls disenchantment. This idea is then expanded by Morris Berman to include a process of reenchantment, which Berman describes in his book, *Reenchantment of the World*. Frederic Jameson hoped to explore a resolution to these issues in his book, *Postmodernism: The Cultural Logic of Late Capitalism*, by developing a new form of spatial and material mapping of our late capitalist/postmodern age. However, Jameson's economic and Marxist philosophies lead into a materialist trap: his concern with the physical appearance of and desire for objects participates in the same process of rationalization which Weber describes, ultimately providing no solution. To avoid such a materialist trap, Wilder's *Our Town* and Miller's *The Man Who Had All the Luck* can be analyzed, using cosmology, to focus not just on social and economic issues, but also moral and spiritual issues that relate to ideas about moral agency, the afterlife, and the passage of time long before and long after human existence. Wilder's play can be interpreted as an idyllic love story about the relationship between George and Emily in the rural New England town of Grover's Corners. However, when analyzed using a cosmological interpretive strategy, *Our Town* reveals human and nonhuman entities whose mutual interactions allow for what can be described as an assemblage of human and cosmic life. Similarly, Arthur Miller's *The Man Who Had All the Luck* can also be analyzed to rebut the materialism presented by Jameson, just as it spins a love story too about the relationship between the characters, David and Hester. David tries his hardest to make a good economic life for himself and Hester; however, he constantly questions the world and his agency as a human when he sees so many people that he knows fail in their endeavors. A discussion of these two plays will focus, first, on using a cosmological interpretive strategy to trace within these works the process of rationalization and disenchantment that these central characters undergo. Second, there will be a focus on how this disenchantment is depicted in each play to reveal that we, as humans, are not predestined to any specific fate according to economic or theological determinism. Instead, the readers are led to realize the theopoetical nature of our lives as interdependent upon each other and the entities in an even wider cosmos. When applied to Wilder's *Our Town* and Miller's *The Man Who Had All the Luck*, this cosmological interpretive strategy reveals that, despite the vastness of cosmological creation unfolding around us and acting upon us, we still retain our own agency. This agency repudiates the economic determinism by which Jameson defines postmodernism and late capitalism. Using this cosmological interpretive framework, it can be seen how both Wilder's and Miller's works reject disenchantment and deterministic thinking and lead readers through the process of reenchantment. Through this process, readers begin to recognize how agency is built around collapsing the binary opposition of the material and immaterial and recognizing a theopoetical, interdependent relationship to cosmological creation.

## 2. Disenchantment, Reenchantment, and Cosmological Ecology

The struggle between a rationally ordered reality and the seeming chaos of an infinitely creative reality is described in Max Weber's analysis of the ascetic and mystical traditions in human culture in *Essays in Sociology*. In the chapter "Religious Rejections of The World and Their Directions", Weber states that, "[i]n our introductory comments, we contrasted, as abnegations of the world, the active asceticism that is a God-willed action of the devout who are God's tools, and, on the other hand, the contemplative possession of the holy, as found in mysticism. Mysticism intends a state of 'possession,' not action, and the individual is not a tool but a 'vessel' of the divine" (Weber 1946, p. 325). Weber describes a categorical difference between asceticism and mysticism in which, under asceticism, the "devout" become the actors, the agents of God's will (God here being used as a term for the cosmic creator and not a depiction of any particular religion). Under mysticism, however, the devout simply recognize that they hold within their corporeal form the divine essence from which everything flows, but which directly rejects that corporeality. Weber states

that both ascetism and mysticism are based on the rationalization and abnegation of the worldly for the otherworldly. Without the split between the corporeal body and the spirit, then ascetism and mysticism could not exist and neither could theories that are predicated on pure materiality, such as Frederic Jameson's philosophies. This trend can be seen in the reasoning behind the creation of economic and social caste systems; the development of purity rituals which reject natural processes or modes of being; the destruction of certain societies from Troy to Jericho, Sodom and Gomorrah, and Carthage; the development of genocide and sectarian violence, such as the Roman purge under Constantine; the rise of the Islamic empires; the European crusades; colonialism and slavery; and eugenics. More recently, society has replaced the idea of God with other terminology that affects the same deterministic destruction, whether it is called science, evolution, the invisible hand of the market, nature, or even democracy when used for colonialized nation-building. Weber calls this process the rationalization of religion (within which he includes magic). Weber recognizes that "the further the rationalization and sublimation of the external and internal possession of—in the widest sense—'things worldly' has progressed, the stronger has the tension on the part of religion become" (Weber 1946, p. 328). It is this pervasive binary opposition of mysticism, ascetism, religion, the divine, or the other worldly on the one hand, to materialism, rationality, science, the profane, and the worldly on the other hand. This binary opposition leads to what Weber defines as "the disenchantment of the world and its transformation into a causal mechanism" (Weber 1946, p. 350). This process of rationalization and disenchantment is also highlighted and used in Wilder's *Our Town* and Miller's *The Man Who Had All the Luck*, which push these categorizations and rationalizations to their limits, while also attempting to counteract their destructiveness. Yet these processes of disenchantment are not often discussed in these plays, due to a lack of theoretical approaches for focusing on such issues—a void which a cosmological interpretive strategy can help fill in order to more fully understand these disenchantments and their ramifications for our society.

Weber goes on to describe how the rationalization of religion first began when society turned from magical ideas about the world, such as animism, shamanism, and certain forms of paganism, to a society centered around religions or religious ideals. Weber writes: "We have said that these modes of behavior, once developed into a methodical way of life, formed the nucleus of asceticism as well as of mysticism, and that they originally grew out of magical presuppositions" that were then systematized "to direct a way of life to the pursuit of a sacred value. Thus understood, the prophecy or commandment means, at least relatively, to systematize and rationalize the way of life, either in particular points or totally" (Weber 1946, p. 327). Weber is explaining the process of taking the idea of God and God's will and putting it into a systematized order in which to build a supposedly coherent and unified society. This requires localizing God's will around a specific group of people at a specific location in a specific time. Wilder's *Our Town* attempts to de-emphasize this localization and specificity (as does Miller's *The Man Who Had All the Luck*), thereby counteracting the process which Weber characterizes as a byproduct of mysticism and ascetism. These plays end up rejecting ascetism altogether and forming a new paradigm for mysticism—one which does not reject the world and its "magic" but upholds it, even pointing us further towards immanence, the cosmological understanding of reality as the entwining of the material and immaterial, where all of creation holds divine essence and is therefore sacred. Morris Berman, building upon Weber's work above in his book *Reenchantment of the World*, refers to this immanent understanding of reality as reenchantment, stating that "the world was enchanted and man saw himself as an integral part of it. The complete reversal of this perception in a mere four hundred years or so has destroyed the continuity of the human experience and the integrity of the human psyche. It has very nearly wrecked the planet as well. The only hope, or so it seems to me, lies in a reenchantment of the world" (Berman [1981] 1996, p. 23). In the sense that, as Berman predicted, "quantum mechanics thus affords us a glimpse of a new participating consciousness" (Berman [1981] 1996, p. 145), this cosmological interpretive framework

can be seen as a continuation of Berman's ideas. However, Berman's focus on process, in particular, individuation (Berman [1981] 1996, p. 78), the symbolism of the process of alchemy (Berman [1981] 1996, p. 108), and the learning process (Berman [1981] 1996, p. 139), describes mostly how over time reenchantment can lead to transformational experiences. However, Berman never sets down any concrete pathway to achieve such experiences, leaving the process shrouded in mystery, which may be his own way to try to fill the gap between the physical and the metaphysical, the worldly and the otherworldly. Instead, it is proposed here that these ideas of disenchantment and reenchantment require us as individuals to take on our own agency, but also to realize that our agency works not in isolation but in concert with natural, economic, cultural, and other cosmic forces. This is similar to what Jane Bennet describes as assemblages in her work *Vibrant Matter: A Political Ecology of Things*. Bennett explains that "Assemblages are not governed by any central head: no one materiality or type of material has sufficient competence to determine consistently the trajectory" (Bennett 2010, p. 24). Instead, Bennett describes how the assemblage is a moving, acting, in this case thinking being, explaining that "Each member and proto-member of the assemblage has a certain vital force, but there is also an effectivity proper to the grouping as such: an agency of the assemblage" (Bennett 2010, p. 24). Ervin Laszlo, known for his work regarding systems theory, comments similarly on this idea of assemblages in his work, *What is Reality?: The New Map of Cosmos, Consciousness, and Existence*, except Laszlo focuses on the relationship between the microscopic and the cosmic. He explains that there is a "new paradigm at the dawn of the twenty-first century" which "sees the world as a whole system where all things interact and together constitute an entangled, quantum -like system in which all things are intrinsic elements in an integral whole" (Laszlo 2016, p. 4). This entanglement, often referred to as quantum entanglement, is based upon what Laszlo describes as the observation "that things may be at any finite distance in space and time but remain nonetheless connected. Such action-at-a-distance is an anomaly for the classical paradigm—even Einstein called it 'spooky'" (Laszlo 2016, p. 4). Laszlo is underscoring the fact that every mode of being in the cosmos is interconnected on a microscopic level and therefore is also then interconnected as a larger system on the macroscopic and cosmic levels. This reveals that the existence of humans depends on these other entities and, at times, their existence depends, and, at the very least, is influenced by human existence. All of our existences, our histories, our stories are interconnected. A cosmological interpretation of Wilder's *Our Town* and Miller's *The Man Who Had All the Luck* reveals these interconnections to the readers. In addition, these texts reveal how our own agency as individuals is a part of the constant flux of innumerable interactions between ourselves, other humans, and nonhumans in a sometimes chaotic, but always creative, cosmos.

### 3. Free Agency within a Cosmic Ecology

Through its use of spatial minimalism in the construction of its stage scenery and atmospheric tone, Thornton Wilder's play, *Our Town*, reveals the struggle between disenchantment and reenchantment. The play was originally performed in 1938, just before the outbreak of World War II. The play begins with these stage directions: "No curtain./No scenery./The audience, arriving, sees an empty stage in half-light" (Wilder [1938] 1985, p. 5). From this description, the world of the play is only partially unfolded, the remaining parts awaiting to be revealed to the audience as they are brought forth into the world by the Stage Manager. It is only after this description of the partially revealed scenery that Wilder begins to give actual physical descriptions such as the town name, "Grover's Corners" (Wilder [1938] 1985, p. 5), and the date of the first act, "7 May 1901" (Wilder [1938] 1985, p. 5). The initial setting, in which everything seems empty, is important because of its attempt to negate a localized time, place, and material reality. This reality is then filled with a specific time and place. However, because of the emptiness of the initial staging, a sense of instability is created in which the scene unfolding could just as easily be any other time and place. Elaine Nelson describes this as a universalizing effect in her essay "The

Universality of Thornton Wilder's *Our Town*," stating that "while retaining costuming and a few sound effects, Wilder abandoned a box set, an involved plot, and highly developed characters" (Nelson 1973, p. 5). Instead, Nelson explains, Wilder "sought universality through the utilization of brief, commonplace scenes, repetition of incidents and phrases, a stage which allows freedom in time and place" (Nelson 1973, p. 5). Nelson here explains how the lack of scenery allows the stage to transcend any localized time or place, creating a mystical and even spiritual atmosphere that, I argue, helps lead the reader toward the path of reenchantment. Further, as much as Wilder presents us with plenty of uncertainty in *Our Town*, he also reveals with certainty the always changing but always present relationship between human existence and cosmological creation. Remember the fact that the stage, at the start, is empty and the lighting set at half-light; the audience is simply told by the Stage Manager that "[t]he time is just before dawn" (Wilder [1938] 1985, p. 5). The Stage Manager also notes that "The morning star always gets wonderful bright the moment before it has to go, - doesn't it?" (Wilder [1938] 1985, p. 6). The Stage Manager not only describes the time as between night and day but, in doing so, also invites the audience to think about the morning star, which is usually associated with beginnings and even with birth and creation. This invitational tactic, possibly borrowed from epic theater, draws the audience into the world of the play, so that they too, play a part in its revelation. Min Shen comments on this section of Wilder's *Our Town* in the article, "'Quite a Moon': The Archetypal Feminine in *Our Town*", and associates the depictions of the moon with fertility and creation as well. Shen writes: "The morning star, also known as Venus, has always suggested the feminine principle because of its association with the goddess of love and beauty. Aphrodite in the Greek pantheon of gods and goddesses. She represents, among other things, passionate sexuality which is linked with fertility" (Shen 2007, p. 7). Fertility, however, is also linked with death, since birth is only able to come about as things naturally mature, getting older and closer to death. Unlike many understandings of death as an end, Wilder shows us in these opening lines that death is part of a larger continuum though which leaving is associated with both beginnings and endings. We have a morning star that is announcing a beginning, a new dawn, while its momentary phase is ending. Wilder here creates an in-between state that defies categorization, becoming not quite a beginning but not quite an ending either: it is just a moment in an eternal continuum of a fertile, creative cosmos filled with a plurality of beginnings and endings, deaths and rebirths. Beginnings and endings are also markers of time passing, and by starting us out in an in-between state that fuses beginnings and endings, Wilder begins to dismantle our notion of time, or at least human time, pushing us toward an understanding of cosmic time. Furthermore, these opening lines also begin to dismantle our notion of space, since the scene is not quite empty, due to the presence of the stage and the audience themselves, but it is not full either, as it is devoid of props. The whole scene, stage, and audience are cast in an ethereal twilight that breaks down the Platonic binaries of dawn and dusk, light and dark, shadowy reflection and actual objective reality; this space is not quite barren but is pregnant with infinite creative possibility and potentiality. Using a cosmological interpretive strategy shows that Wilder's artistic creation begins in an eerie, nascent state, pushing the audience past notions of endings and beginnings and toward the realization that the cosmos is always in eternal flux. Wilder pushes the audience toward an understanding of the cosmos that shows its mystical nature—a nature that defies categorization but not understanding. Wilder moves the audience past disenchantment toward reenchantment, reminding the audience of the mystical nature of cosmic creation.

This mystical atmosphere is later revealed to be at odds with certain key facts that shape the characters' lives. For instance, the Stage Manager comments on the tragedy of war for Joe Cromwell, who was "[g]oin' to be a great engineer, Joe was. But the war broke out and he died in France—All that education for nothing" (Wilder [1938] 1985, p. 10). The Stage Manager's comment is very jaded and claims that Joe Cromwell had the potential to benefit society and the cosmos. Joe's profession was to be an engineer, and, as engineers are creators, they build from our natural resources the architectural structures that are

then used by society; however, following Joe's death his education is "for nothing." The Stage Manager's jaded tone reflects the disenchantment many in society felt after World War I. Almost immediately after Joe's death is revealed, however, the audience is told that Dr. Gibbs is returning home after being up all night delivering "twins born over in Polish Town" (ibid., p. 9) and that Joe's teacher is "getting married to a fella over in Concord" (Wilder [1938] 1985, p. 9)—events traditionally associated with themes of marriage and new life. Wilder demonstrates to the audience how life may end for some but that the cosmos is generative; it keeps creating new life, in part through the interactions we as humans have with each other through marriage and sexual relations. Further, Dr. Gibbs asks Joe about his knee, implying that Joe has had some type of injury to it. Joe replies that his knee is fine but that, as Dr. Gibbs told him, "it always tells me when it's going to rain" (Wilder [1938] 1985, p. 9). This comment may sound innocuous to a scientistic worldview, and Joe's power to predict might be explainable, in this view, by a change in air pressure acting on bodies. A cosmological perspective, however, would want to indicate that the point goes beyond this superficial understanding and that the molecules are acting, not just on us but in us, in our very bones, and in the very molecules that make up our bones. Wilder demonstrates how our lives, our flesh and bone, and our very molecules are intricately linked to our natural surroundings. These surroundings have their own existences, their own actions and agency, their own time, and even their own history, just as we do.

This intertwined existence we share in this way with nonhuman existences is, as Lazlo called it, our quantum entanglement. Bruno Latour also comments on this quantum entanglement in his book *We Have Never Been Modern*. Latour explains how actors, from Copernicus to Boyle, have based their scientific ideas around the idea that humans are separate from nature but that they did not consider the connection between our molecular composition and the molecular composition of the objects and elements we interact with. Latour states: "[t]here was indeed a contingent history, but for humans alone, detached from the necessity of natural things" (Latour [1991] 1993, p. 1672); instead of this contingent history, he argues that "each entity is an event" (Latour [1991] 1993, p. 1663) in time and history. If we then "redistribute essence," or the ability to act according to our own will, "to all the entities that make up this history" (Latour [1991] 1993, p. 1667) then these entities "become mediators—that is, actors endowed with the capacity to translate what they transport, to redefine it, redeploy it, and also to betray it" (Latour [1991] 1993, p. 1667). Further, Latour argues that this redistribution of essence transforms our understanding of time and history such that "[a]ll the essences become events, the air's spring by the same token as the death of Cherubino. History is no longer simply the history of people, it becomes the history of things as well" (Latour [1991] 1993, p. 1677). Latour's flourish in this comment, linking spring, a time of marriage and rebirth, with the death of Cherubino, the young page from Mozart's opera, The Marriage of Figaro, reflects the situation at the opening of *Our Town*. As we have seen, this play begins by linking the spring of a new century, May 1901, with the birth of the twins, the teacher's marriage, and Joe Cromwell's looming death in World War I. When this context is considered, the comments of Joe and the Stage Manager are incredibly powerful as an index of the interconnected relationship between new life and future death. Wilder's *Our Town*, through the comments of Joe and the Stage Manager, advances a step toward the process of reenchantment, reminding the audience of the passage of time and how short our time is here on Earth. The play also reminds us how many atmospheric, molecular, and social interactions it takes from so many entities in the vast history of the cosmos to bring our lives into being and to let us act them out for just a short passage of time.

Furthermore, if our existence is a product of the interactions of multiple entities, then our agency also derives, in part, from those interactions. This view rebuts the theological and divine determinism of the old ascetics and mystics, as described by Weber. Such determinism is not confined to theology or asceticism; however, economic determinism is at the heart of the Marxist thinking, to which theorists like Fredric Jameson subscribe. According to Jameson, late capitalism is predicated on the subjugation and suffering of

others. Jameson writes that the goal of his work, *Postmodernism, or the Cultural Logic of Late Capital*, is to trace "a protohistorical narrative in which something is affirmed about the specificity of this particular period, including its waning and its imminent transformation into something else—this narrative is most clearly grasped in economic terms" (Jameson [1984] 2003, p. 192). Jameson does admit that postmodernism can be defined as a literary period, but he feels that the best means of distinguishing postmodernism from what has come before is to attend to the transformations of late or multinational capitalism, which, in his view, radically problematizes the ability of literature to contest its ideology and provide a counterpoint to economic determinism. Of course, Jameson also believes that economic determinism is nothing new; it is taken for granted in the Marxist methodology which he employs to diagnose it; however, Jameson believes that this economic determinism has become something sublime in postmodernism. Throughout the play, *The Man Who Had All the Luck*, the character Dave and all of his friends all act as if their economic fate is already predetermined. However, to the chagrin of many critics who have bought into Marxist philosophies, the play ultimately ends up rejecting such economic determinism.

Just as the opening of *Our Town* reveals and addresses the disenchantment caused by war, Arthur Miller's *The Man Who Had All the Luck* addresses both theological determinism and economic determinism in ways that counteract, at times, Jameson's view of global or multinational capital. In the introduction to the Penguin Classic version of the play, Christopher Bigsby notes that Miller's *The Man Who Had All the Luck* premiered "at the Forest Theater in New York on 23 November 1944", (Miller viii). One could argue that Miller's play is a bridge between the modern and the postmodern, debuting just a few months before the end of World War II, which is often identified as the starting point of the postmodern age. Many critics, such as the ones listed above, focus on the economic issues within *The Man Who Had All the Luck* and Miller's other works, such as *Death of a Salesman*, seeing them solely as critiques of capitalist culture. In the play, David Beeves seems to find success everywhere he turns. However, he feels sometimes he does not deserve this success and, at other times, he feels that the unfortunate experiences of others are not deserved. David begins to become disenchanted, questioning why such troubles and horrible events must occur and who is responsible for such events. He also begins to question whether these events are caused by human agency or through some deterministic forces outside the realm of the human. This line of questioning leads David to the point of paranoia, fearing that some catastrophe will be sent his way to counter all of his successes. This fear is often seen as guilt toward capitalist culture. Jane Dominik addresses some of these issues in her compilation of the theatrical reviews of Arthur Miller's works, aptly entitled "Critical Receptions of Arthur Miller's Works." Dominik reduces the nature of the critics' reviews to Miller's themes of "guilt, responsibility, illusions, dreams, family, betrayal, the "birds coming home to roost" (Bigsby 2017, Company 49), and "success and failure in capitalistic America" (Dominik 2010, p. 72). Other critics focus on the breadth of Miller's topics and his dramaturgical style, for instance, Dominik quotes critic George Creeley's analysis of Miller's slice of life rendering of *The Man Who Had All the Luck*, stating specifically that Miller's deficiency was "to focus his plot and his characters so that a clear dramatic image will be created" (as quoted in Dominik 2010, p. 74). Creeley felt that the play is ambiguous as to what its overall message should be. This focus on the practical dramaturgy of *The Man Who Had All of the Luck* is what seems to really pervade the criticisms of the play. Martin Gottfried chronicles such a critical reception in his book *Arthur Miller: His Life and Work*. Gottfried quotes Lewis Nichols of the *New York Times*, stating about the play's debut that he saw only "'one or two effective moments' amid 'the confusion of the script [and] its somewhat jumbled philosophies'" (Gottfried 2003, p. 81). Gottfried himself explains what many of the critics saw as an unbelievable ending, stating that "The play finally expires of too many twists. An Arthur Miller who is unable or unwilling to decide whether he wants David to commit suicide straddles the issue" (Gottfried 2003, p. 80). Miller himself comments on this issue concerning the ending of the play, stating about the advice one

critic gave him, "the critic for the Hearst paper, *The Journal-American*, asked me to meet him at the New York Athletic Club. He was the first critic I'd ever laid eyes on, a good-looking guy named Anderson. He said, 'That play was a failure, but the mistake you made was not to make it a tragedy'" (Guernsey and Miller [1987] 2015, p. 37). The critics were stunned by the fact that fate does not come for David Beeves and give him his catastrophe. These critics essentially accept David's worldview, which the ending rejects. David believes that he is not responsible for his success, but that either fate, destiny, or divine providence is instead. However, a cosmological interpretation reveals how the ending rejects this premise, showing that David is responsible for his success. Unbeknownst to him, David is given feed for his mink that has been tainted with worms. David combs through the feed, meticulously throwing away any spotted or discolored feed, disposing of the infected pieces without even knowing that the batch of feed has been tainted. Though reacting to a situation that originally was beyond his control, the distribution of the tainted feed, David becomes the author of his own fate, ensuring his animals are safe and succeeding in saving his family from ruin, due to his own hard work and diligence.

This discussion about David's fate and the ideas about determinism that it brings up is first mentioned in the play by David's friend, Shory. Shory raises the issue when he remarks "A man is a jellyfish. The tide goes in and the tide goes out. About what happens to him, a man has very little to say" (Miller [1994] 2004, p. 19). Shory seems to be implying that there is a determining force which holds absolute power over our actions and fate. David seems convinced and agrees with Shory. This questioning of the human capacity for agency continues further, as David asks about his own success, relative to his brother Amos's failures: "Why? Is it all luck? Is that what it is?" (Miller [1994] 2004, p. 48). David goes on to ask, "Am I that good and he that bad?" to which he then exclaims, "I can't believe it. There's something wrong, there's something wrong!" (Miller [1994] 2004, p. 49). Further emphasizing the anxiety about a predetermined reality, David here seems to believe that there is some cosmic conspiracy that is producing his good fortune and his brother's misfortune. In contrast to David's developing sense of determinism, however, other characters in Miller's play clearly reject the notion of a predetermined cosmos. The baseball scout Auggie Belfast explains to Amos and Pat, Amos's father, that it wasn't luck or fate or God behind the fact that Amos doesn't get selected to the baseball draft. Instead, it is a result of Amos's inability to concentrate on the ball field: "as soon as a man gets on base and starts rubbin' his spikes in the dirt and makin' noise behind your boy's back, something happens to him . . . your body, Mr. Beeves is floating somewhere out in paradise" (Miller [1994] 2004, p. 53). In Auggie's view, Amos's lack of concentration is not a cosmic mystery, but a result of the fact that Pat and Amos have been training in their cellar since Amos was nine years old. Auggie tells Pat that "[i]n the cellar there is no crowd. In the cellar he knows exactly what's behind his back. In the cellar, in toto, your boy is home" (Miller [1994] 2004, p. 54). Auggie goes on to explain that once Amos "gets out on a wide ball field, and a crowd is yelling in his ears, and there's two or three men on bases jumpin' back and forth behind him, his mind has got to do a lot of things at once, he's in a strange place, he gets panicky" (Miller [1994] 2004, p. 54). So, it is not luck, fate, or God that determines Amos's failure but, instead, Amos and Pat's decision to practice only in their cellar, rather than outside or on an actual ball field. The cellar's effect on Amos is that it trains Amos to focus only in the quiet solitude, so that once he is out in the noise of the rest of reality, he cannot focus his attention. This is not a predetermined outcome but a direct result of the agency of Amos, Pat and their interactions with their surroundings. David's disenchantment results from his refusal or inability to recognize that he, Amos, and Pat are in part responsible for their own agency in connection with their interactions with other entities.

Even with her steadfast belief in Miller's works as simply critiques of capitalism, Lois Tyson observes a similar refusal to recognize agency in Miller's *Death of a Salesman*. Willy Loman does not accept his responsibility for contributing to the cut-throat and immoral atmosphere that leads to his own failures. Tyson explains in her essay, "The

Psychological Politics of the American Dream: *Death of a Salesman* and the Case for an Existential Dialectics", that "Willy's failure to see the obvious unscrupulous underside of Ben's financial success, like the rest of his apparent moral confusion concerning his and his sons' success-oriented ethics, is not the result of innocence or ignorance, but of selective perception" (Tyson 1992, p. 264). Tyson argues that Willy's success or failure is the result of how he has perceived and reacted to the opportunities and examples around him, including adopting unethical practices and behaviors. Willy is, in part, the agent of his own actions, his own downfall, and is not simply the victim of the deterministic economic conditions of capitalism. Tyson goes on to question how agency in literature has traditionally been interpreted, stating that "[o]nce we begin to see the ways in which the individual subject is neither wholly an autonomous agent nor merely a social product, the conceptual space thereby opened makes room" (Tyson 1992, p. 261). Tyson elaborates on this point, arguing not for "a return to the autonomous subject the ethical critics want to construct, but for a return to and dialectical reformulation of the existential subject" which has largely been "neglected since the advent of post-structuralism" (Tyson 1992, p. 261). Tyson here reminds us that emphasizing agency does not mean taking a stroll down a nostalgic memory lane but, instead, re-evaluating our current perceptions and conceptions of agency and determinism, particularly when the current theoretical models do not seem to fit the social circumstances. Just as Miller, according to Tyson, uses Willy Loman in *Death of a Salesman* to explore these questions of agency, so too does Miller use David in *The Man Who Had All the Luck*. David's disenchantment is a result of his lack of understanding of his agency and the agency of others. By bringing attention to the question of agency, *The Man Who Had All the Luck* leads the reader toward a process of reenchantment. Whether we appreciate all that the cosmos provides, whether we are good stewards of what is provided, and whether we recognize our own agency in our lives and the forces that also influence that agency are central issues informing both Miller's and Wilder's plays.

## 4. Theopoetic Cosmology

Within the context of a discussion of agency, it is important to remember that the title of Miller's play is *The Man Who Had All the Luck*. Use of the word "man" rather than "Dave" in this title is significant in that, like the title *Our Town*, no specific person, time, or place is evoked; instead, the audience or reader is asked to think about the general conditions that might make any "man" lucky. Nor is it a coincidence that the two titles and themes evoked by them are so similar; on the contrary, Miller studied Wilder's work, and was deeply influenced by *Our Town* in particular. Stephen Marino, in his essay "Thornton Wilder and Arthur Miller" reminds us that Miller was fond of conversing with Wilder and writing about his works. Marino discusses Wilder's influence on Miller's writing, stating that Miller was very much concerned with *Our Town*, praising it as a "poetic drama" as opposed to a "realistic' play" (Marino 2013, p. 1403). Marino explains how Miller saw that "the entire play is occupied with what the title implies: the town and the society, and not the family" (Marino 2013, pp. 1420–23). The title itself, *Our Town*, is a poetic device representing a set of interconnecting relationships; in this manner, it breaks away from modernist objectivism by refusing to just say "the town", just as it also breaks away from the individualism of the first person singular possessive represented by "my town," favoring instead the more communal and universal *Our Town*. As a communal phrase, "Our Town" also de-emphasizes any particular time and place in favor of the consideration of the town as a collective series of relationships. Similarly, Arthur Miller styled his title, *The Man Who Had All the Luck*, in a way that could refer to anyone at any time. This human being is the one who has all of the luck; but luck is simply good fortune and good fortune is another way of saying a person who has a good fate. The question which Miller concerns himself with is, who is responsible for doling out such fortune/fate: is it God, the economy, or some combination of these with the possibility of human agency? The titles of *Our Town* and *The Man Who Had All the Luck* become prompts to engage in cosmological questioning about human agency. These universalized titles deny the localized time and place; furthermore,

as the plays themselves unfold, the time and place within the text keeps switching. This constant flux of time and place within both of the plays throws into doubt the ideas of either divine determinism or Jameson's economic determinism.

One major issue with using Fredric Jameson's ideas (or any other theory based on the stratification of classes or hierarches) to interpret these narratives is that they fall directly into the trap of objectivism and scientism, where only the materially observable reality is of use or importance. Take, for instance, Jameson's attempt to recapture the sublime of history in his 2003 essay "Future City", in which he details the history and contemporary status of the shopping mall, one of the prime examples for Jameson of the deterministic monotony of late capitalism. Jameson writes that "[t]he problem to be solved is that of breaking out of the windless present of the postmodern back into real historical time, and a history of human beings" (Jameson 2003, p. 5). Jameson goes on to say that he intends for the essay to be a "dystopian appearance" that acts as a "sharp edge inserted into the seamless Moebius strip of late capitalism" (Jameson 2003, p. 5). Jameson admits though that even he, at times, can find "trajectories with their magical moments" "that have pseudo-temporality of matter ceaselessly mutating all around, moments of rare, breathtaking beauty" but that such moments are "scarcely enough to compensate for the nightmare" ("Future City" 4). This is economic and philosophical materialism at its worst. Despite wishing for some magic to reenchant his existence and compensate for the existence of the used and reused "Junkspace" (Jameson 2003, p. 3), as he calls it, Jameson is only concerned with our spatial existence within the confines of the flow of material wealth within our social and economic power structures. By limiting his discussion to only these material concerns, Jameson gives these structures a totalizing and deterministic narrative power over our lives and, as he says, over the course of human history itself. This has been the problem historically with many of our philosophical, theological, and teleological narratives, that have tried to be the only narrative about reality. Now, all of our social and economic structures, secular and religious, have come into question. William Beardslee reacts to Jameson's nightmarish materialist world in his article "Stories in the Postmodern World: Orienting and Disorienting", from the collection of essays, *Sacred Interconnections: Postmodern Spirituality, Political Economy, and Art*, which was edited by David Ray Griffin (1997). Beardslee concludes in his essay that:

> "If our own stories are created by the interaction of data from the past with continual purposive events of unification of experience, then narrative is far more than the arbitrary play it often seems to be in literary postmodernism. Stories are indeed an interweaving of many strands, and not the single story which they often seem to be. A principal task of postmodern imagination is to help us see ourselves in this more complex way. To do so, we still need to relate ourselves to overarching stories. These stories cannot simply be invented. We are, as the French are fond of saying, "bricoleurs," tinkerers, cobbling together a structure out of rather miscellaneous elements given to us by our pasts. But the elements from the past are not mere unrelated fragments, despite their miscellaneous character. They offer us tracts of meaning, directional, transformative possibilities as we relate our own stories to them". (Beardslee 1997, p. 172)

Beardslee and Jameson agree on the importance of our ability to link our narratives together to find some sense of identity and purpose. One of the major differences between them, though, is with regard to scale. Jameson is specifically talking about a "globalization" (Jameson 2003, p. 2) of materialism and consumerism.This "globalization," as large as it seems when compared to our tiny individual selves, is small when compared to the overarching reality of our universe. Furthermore, our own universe is even theorized by quantum mechanics to be only one within a much wider multiverse that reveals and reminds us of the fact, as Beardslee is supposing, that we are a part of a truly complex narrative of cosmological creation. Both Miller and Wilder attempt to reveal this plurality of worlds in their texts.

Just as Jameson is concerned with the capitalistic determinants of the postmodern world, Wilder and Miller both also devote a narrative layer to the social and economic power

structures critiqued under modernism and postmodernism. Both stories are also concerned with humanity's relationship with an existence beyond the control of these social and economic structures. In Wilder's *Our Town* and Miller's *The Man Who Had All the Luck*, the characters are constantly having to navigate a rapidly changing world and landscape that exceeds economic determination. Wilder and Miller fill their stories with happy moments, along with moments revealing disenchantment, brokenness, and suffering, as Beardslee puts it, that are essential to human experience. Rather than being focused merely with spatial and material concerns, which for Jameson define most of postmodernist literature, I argue, instead, that much of modernist and postmodernist literature is concerned with questions about the cosmic order. There are many implied cosmic questions within both *Our Town* and *The Man Who Had All the Luck*. One of these implied cosmic questions emerges when Miller's character J.B. comments about his infertility issues: "Yeh, no kids. Too old. Big, nice store with thirty-one different departments. Beautiful house. No kids. Isn't that something? You die, and they wipe your name off the mailbox and . . . and that's the ball game" (Miller [1994] 2004, p. 4). This question is not mere barstool rambling but reveals the disenchantment felt by the populations of Wilder and Miller's time concerning theological and scientific depictions of a vast and mechanistic world they have become trapped within—a world governed by random "luck" or divine fate. If luck is dished out exclusively by these forces or others like them, say economics, history, genetics, and biology, then our reality is completely predetermined for us. Basarab Nicolescu, in his book *From Modernity to Cosmodernity*, explores the idea of luck, fate, and chance, explaining "The word chance corresponds to the word hazard in English. In turn, the word comes from the Arab az-zahr, which signifies "play of the dice" (Nicolescu 2014, p. 43). Nicolescu explains further that "the quantum event" was seen as "an accidental event, owing to a play of the dice (played by whom?)" (Nicolescu 2014, p. 43). Nicolescu is reminding us here that in classical understanding even the existence of matter and particles was seen as chance, as someone playing a game or toying with reality. The question of who was playing the game has changed from time to time, shifting from God to nature, economics, biology, history, or culture. As Nicolescu notes, however, "Quantum randomness is really a constructive gamble, which has a meaning—that of the construction of our own macrophysical world. A finer material penetrates a grosser material. The two coexist and cooperate in a unity that extends from the quanton to the cosmos" (Nicolescu 2014, p. 43). Nicolescu reveals that our reality is not predetermined by chance, luck, or cold probability, but by our actions interacting on several levels with the actions of the other existing entities in the cosmic and quantum world. This opens reality to an infinite series of outcomes and potentialities. Miller too is tapping into this acknowledgement, in order to combat the economic, divine, and cosmic determinism of his day.

Furthermore, in the essay "Arthur Miller and the Art of the Possible", Steven Centola argues that Miller recognizes that the possibilities inherent within the whole dramatic event are "limitless" (Centola 2005, p. 64). Centola goes on to say that, for Miller, "the fundamental indeterminacy of meaning—an indeterminacy that Roland Barthes says inevitably results from the plural nature of the play text as a discourse that can be experienced only in the art of production— poses no nihilistic threat in Miller's world" (Centola 2005, p. 64–65). Centola recognizes that Miller's plays reveal not a predetermined result but an openness to varied and rich experiences, even when these plays end in tragedy. In Centola's view, "the theater is a place where nature is transmuted into art, where reality meets and fuses with illusion, where text and subtext, character and action, word and gesture become one, where opposites are held in balanced suspension, and that, of course, is why the theater is the realm of the possible" (Centola 2005, p. 66). Centola's analysis helps us to see that, by raising these questions about the fairness and agency responsible for life's outcomes, Miller and Wilder reflect the anxieties their audiences had about the nature of human existence and cosmological creation and assure them that despite tragedy, one's results in life are not foregone conclusions but the collection of innumerable opportunities and choices that one makes as an individual in relation to others and to the cosmos. In turn, *Our Town* and

*The Man Who Had All the Luck* lead the reader toward the process of reenchantment by reminding the reader of their own agency, and how that agency is intertwined with the agency of other entities in the cosmos.

Wilder gives more examples of this plurality of worlds through the letter which Rebecca tells George that her friend Jane Crofut "got from her minister when she was sick" (Wilder [1938] 1985, p. 45). Rebecca goes on to say that the envelope was addressed to: "Jane Crofut; The Crofut Farm; Grover's Corners; Sutton County; New Hampshire; United States of America" (Wilder [1938] 1985, p. 45). When George questions why that is so important, Rebecca replies that "it's not finished: the United States of America; Continent of North America; Western Hemisphere; the Earth; the Solar System; the Universe; the Mind of God" (Wilder [1938] 1985, p. 45). Both Wilder's comments and Nicolescu's comments about the finer and grosser materials show how our physical location reveals that we start with someone seemingly so small, Jane Crofut, who is located within a larger physical structure, her farm, but then that is located within a larger relational structure, the town of Grover's Corners. Further, this expands to the larger physical structures, such as the continent and the Earth. However, as we start to spread out further, we get into finer and finer distinctions, such as the Solar System which is not an actual physical structure but, in the same way as a town, is a relationship between different entities within a given area of a larger relational structure, the Universe. Finally, Wilder widens the interconnections of these relationships until the moment of revelation, realizing that all of existence is located within the Mind of God as a part of the larger cosmic being. As humans, we can see our brain as a physical structure, but we cannot see our mind as a location of consciousness. Similarly, Wilder here configures a theopoetical understanding of reality. Theopoiesis literally means "god creating" and suggests that the basis of our newly expanding perception of agency reveals that we are influenced by the creative forces of culture and the cosmos, but that we too are influencing the future creative outcomes of our culture and the cosmos, including our concept of God. Wilder unveils this theopoetical conception of God in which the structures of our material reality are seen as the physical manifestations of the mind and the consciousness of God, which would mean that God's consciousness, God's divine essence, is interwoven into every physical aspect of our reality. A cosmological interpretive strategy reveals how our material reality is filled with a host of physical entities and as we, as humans, interact with all of these physical aspects and entities, we enact our quantum entanglement, as described by Latour and Nicolescu. We act on these physical entities and they act on us. If God is woven throughout our physical existence in us, in our flesh and bones, in our molecules and we are acting with God and on God, then we are enacting a theopoetical existence with God, creating reality as we go. By using a cosmological interpretive strategy, it can be shown how Wilder's *Our Town* leads the reader toward the process of reenchantment by revealing to the reader the theopoetical nature of our reality.

Miller also reveals a theopoetical concept of reality in *The Man Who Had All the Luck*. Dave becomes upset when, towards the end of the play, while dabbling in mink farming, he double-checks the purity of the animal feed and finds tainted feed, but forgets to call his neighbor and mentor, Mr. Dibble. Dibble, who fails to see the tainted animal feed, is ruined, leaving Dave horrified again that the universe has determined that he should succeed, while others experience misfortune. Dave's wife Hester and his friend Gus, however, question this deterministic conspiracy theory. Hester states that Dave's ability to avert disaster by throwing the tainted feed away "wasn't something from the sky, dear. This was you only. You must see that now, don't you" (Miller [1994] 2004, p. 82). Hester tries to remind Dave that it was his attentiveness to his farming and his hard work in sorting through the feed that allowed him to avert this disaster. Dave's friend, Gus, echoes this sentiment, stating that "[o]f course bad things must happen. And you can't help it when God drops the other shoe. But whether you lay there or get up again—that's the part that's entirely up to you, that's for sure" (Miller [1994] 2004, p. 83). Gus explains to Dave that, despite the challenges he comes up against, where others take success for granted, relying on faith or cosmic providence, it is Dave himself who does the hard work to solve

those challenges through double-checking his feed and avoiding catastrophe. Dave's only problem is that he has resented his success, rather than embracing it and appreciating it, so Gus finally tells Dave to "grin and bear it" (Miller [1994] 2004, p. 83). Gus's comments here reveal how our existence is intertwined with the actions of other entities; however, it is our ability to react to those interactions that allows us to move forward and not be paralyzed by a situation. To act and react in this way is to explore the full extent of our agency. In acting, we then cause our own luck, our own fortune, pulling forth and unfolding the reality of our existence in that moment. In this sense, Miller's *The Man Who Had All the Luck* also serves as a literary form of reenchantment, by revealing a theopoetical concept of the cosmos and of God, not as a concept of some puppet master removed from reality, but as a force we encounter and interact with at every moment.

Wilder doesn't confine his theopoetical conception of agency to just this one section of the play. Just as this concept suggests, Wilder reveals a path toward this theopoetical concept of God by weaving discussions throughout *Our Town* about time and the eternal— something which is beyond time and is a term generally used to refer to God's essence. Even though the play was originally written and performed in 1938, at the beginning of the play, the Stage Manager tells the audience that the "day is 7 May 1901" (Wilder [1938] 1985, p. 5) and that the "[f]irst automobile's going to come along in about five years" (Wilder [1938] 1985, p. 6). The Stage Manager uses the future tense, but he is not telling a prophecy here, but instead is putting the audience in two sequential times at once. The Stage Manager does this again when he asks Professor Willard to come and give details about "our past history here" (Wilder [1938] 1985, p. 21). Professor Willard reveals that the town "lies on the old Pleistocene granite of the Appalachian range. I may say it's some of the oldest land in the world. We're very proud of that" (Wilder [1938] 1985, p. 21). Professor Willard adds that there is also "a shelf of Devonian basalt" and "vestiges of Mesozoic shale, and some sandstone outcroppings; but that's all more recent: two hundred, three hundred million years old" (Wilder [1938] 1985, p. 21). These facts are then contrasted with the Stage Manager's request for some words "on the history of man here" to which Professor Willard replies by referring to "Early Amerindian stock" that, unfortunately, is "now entirely disappeared" except for "possible traces in three families" (Wilder [1938] 1985, pp. 20–21). Despite the possible disappearance of this Native American stock, there is a possibility that it has continued in the blood of the new migrants who are "English" and "Slav and Mediterranean." Even more interesting is that, when the Stage Manager asks for the population of "*Our Town*," he is told by Professor Willard that it is "2640" but is then corrected by the Stage Manager's whisper to "2642" (Wilder [1938] 1985, p. 21). This revised number recalls the first pages of the play, when the audience is told about "some twins born over in Polish Town" (Wilder [1938] 1985, p. 9). Wilder here is pushing us back and forth between the past, present, and future, forcing us to reconsider our concept of time. Wilder reminds the audience that the now is not isolated but part of an infinite continuum.

This concept of the now as part of an infinite continuum is, of course, not unique to Wilder. In Amy Elias's edited collection *Time: A Vocabulary of the Present* (Burges and Elias 2016), Heather Houser explains in her article "Human/Planetary" (Houser 2016) that there are several layers to time. There is human time, based on "human biological" rhythms, and nonhuman or geological time, that works on "planetary rhythms"; according to Houser, these two types of time "partially harmonize around seasonality" (Houser 2016, p. 144). Houser goes on to describe a third type of time, the "inhuman time of instrumentation, computation, and mathematicization that mediates the entrenched binary of human and planetary time" (Houser 2016, p. 144). Houser further explains that using geologic time is important because it "evokes a past whose residues await discovery in the present; it thus captures the then, the 'now,' and what carries 'on' into the future" (Houser 2016, p. 144). To put this in perspective, Wilder gives us an imaginary character, the Stage Manager, who reminds the audience of these various types of time, creating a "now" in 1901 that also looks ahead to known events in 1906, when the first automobile will arrive. This Stage Manager, with the help of the Professor, then throws us back millions, if not billions,

of years into geologic time, showing us all of the processes that it took just to allow the conditions for those human twins to exist. Using a cosmological interpretive framework in this way, one can see how Wilder goes beyond Jameson's and even Weber's accounts of human history and rationality to a cosmic notion of time that details parallel human and non-human histories and the relationship between human and non-human existence that takes place as a result. A cosmological interpretive framework helps to dismantle our common notions of time, to call attention to this relationship between human and non-human existence. This use of an ephemeral and geological setting is one method by which Wilder attempts, through an act of imagination and drama, to lead audiences away from materialistic concerns toward a cosmic unknown, in which all of the things which we might imagine are potentially possible. In the introduction to his collection *Three Plays: Our Town, The Skin of Our Teeth, The Matchmaker*, Wilder remarks that nineteenth and early twentieth century playwrights "loaded the stage with specific objects, because every concrete object on the stage fixes and narrows the action to one moment in time and place" (xi). Wilder saw this as antithetical to the power of the theatre because "as an artist (or listener or beholder) which truth do you prefer—that of the isolated occasion, or that which includes and resumes the innumerable?" (x; author's parentheses and quotes). Wilder answers that "The theatre is admirably fitted to tell both truths. It has one foot planted firmly in the particular, since each actor before us (even when he wears a mask!) is indubitably a living, breathing 'one'; yet it tends and strains to exhibit a general truth since its relation to a specific 'realistic' truth is confused and undermined by the fact that it is an accumulation of untruths, pretenses, and fiction" (xi; author's parentheses and quotes). It is in attempting to show this plurality of truths that Wilder juxtaposes the lives of humans with the surrounding territory. This plurality of truths is seen again when Wilder's character Emily is depicted in the afterlife, desiring to visit the land of the living just one more time.

After marrying her childhood sweetheart, George, and dying during the birth of her first child, Emily is depicted in the afterlife re-examining her earthly life. After seeing how her family members barely paid attention to one another while they lived, Emily cries out: "Oh, earth, you're too wonderful for anybody to realize you" (Wilder [1938] 1985, p. 100). Emily then goes on to ask: "Do any human beings ever realize life while they live it?—every, every minute?" to which the Stage Manager replies "No", but then, after a pause, says "[t]he saints and poets, maybe—they do some" (Wilder [1938] 1985, p. 100). This all leads the reader to open their minds to the third act, which takes place while Emily exists simultaneously in the spirit world and the physical world, reliving a past life that is also the present for those she is witnessing. This overlapping of realities leads Emily to understand that there is always so much "going on and we never noticed" (ibid., p. 100). Wilder has stated that his play is not a "speculation about the conditions of life after death" (xii); instead, I argue that Wilder is most concerned with allowing humans to recognize the beauty of life through revealing that death is not the end but just a transformation into another phase of a mysterious yet infinitely creative universe. It is Emily's recognition of this transformation that is key. This is discussed in critic Kristin Bennett's essay "The Tragic Heroine: An Intertextual Study of Thornton Wilder's Women." Bennett explains how Wilder presented "individuals who unconsciously perform behaviors that have been rehearsed or abided by, over time", lending his works "to analysis through the lens of Judith Butler's theories of gender as performance" revealing how "the socialized repetition that inhibits humans from existing as unfettered agents within their own lives" (Bennett 2012, p. 1175). This is what Emily laments when she decries how no one ever realizes life as they live it. The Stage Manager's answer to Emily here helps to explain how art, and recognizing one's interdependence on humans and nature, can lead to a more conscientious way of living. The Stage Manager identifies poets as the cosmically holy and anointed ones who roam the earth. The poet watches and observes the cosmic beauty of the world and puts this beauty into language. This is an act of creation. In *Our Town*, this act of creation not only allows the poet to appreciate and better understand the wonder of cosmic

creation but to enact it in miniature through the creation of his art, which attempts to render the cosmic creation into language for the understanding of the masses. It is in this respect, as translators of the beauty of cosmic creation, that Wilder places the poets, writers, and artists alongside those divinely chosen speakers of cosmic creation, the saints. The play then becomes the impetus for us to question our material reality and what some call the theopoetical nature of the cosmos. The theopoetical aspects of *Our Town* lead the audience toward the process of reenchantment by making the audience become aware of the innumerable possibilities and potentialities of creation within the cosmos—and of our agency within that cosmos.

## 5. Conclusions

The questions about space, time, place, and agency brought up through a cosmological interpretive approach to *Our Town* and *The Man Who Had All the Luck* reveal a shift in the literary representations of our relationship to the cosmos, such as from economic and divine determinism to a theopoetical understanding of the agency of the individual. This shift moves the reader away from a deterministic view of agency to a more theopoetical perception of agency. In *The Man Who Had All the Luck*, Dave accepts the help of Gus and becomes a successful mechanic. Dave then becomes a successful mink farmer by thoroughly checking his feed every day. Dave also takes his own health and the health of his wife seriously, and is thus able to have a healthy child. Dave has not been the sole reason behind his success, but he is one of the agents most responsible for it. Similarly, while a ghost reliving the past, Emily in *Our Town* at first despairs when viewing the choices she made in life, but then recognizes that those same choices created the opportunity for an incredibly numerous array of meaningful experiences in human and cosmic life. As Emily points out, we as humans sometimes just shut ourselves in our little boxes; however, it is entirely within our own power and our own agency to engage with the multiple possibilities of the present to make our lifetime a truly rich experience.

**Funding:** This research received no external funding.

**Acknowledgments:** Thank you to Alex Blazer, Christopher Kocela, Matthew Roudane, and Paul Schmidt for your encouragement and support!

**Conflicts of Interest:** The author declares no conflict of interest.

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
