# Peer review of "Thornton Wilder and Arthur Miller: A Brief History of Time, Space, and Matter"

_2410-9789, doi:10.3390/literature2030010_

Round 1

Reviewer 1 Report

Dear author, a comparison between two seminal dramatists of the American literary canon is well taken, but must be well executed. Scholarship on both Wilder and Miller is extensive, so in order to contribute and advance our understanding, the argument must be original and well-honed. I am afraid it is unclear, however, what you want to accomplish: is this truly meant as a literary studies paper or as a philosophical tract that is a somewhat subjective bashing of materialism? The major flaw in my view is that the primary source analysis is successively neglected in favor of a lot of secondary and theoretical literature (which is in many instances unnecessary to the extent it is done). This results in the fact that you are writing about drama, two American plays, without engaging in the various kinds of theater we see in the first half of the 20th century. In the present case, the term "epic drama" does not appear once, even though it would cover most of your arguments and interpretations. In turn, though you are after a specific addition to scholarly debate, which you call "cosmological interpretive strategy," it is not so original after all, and neither so very clear. In my opinion, this paper needs a thorough re-write after a solid engagement with realist / epic / minimalist theater. 

Author Response

    Thank you for your critique! I have attempted to make all of the grammatical and syntactic changes requested. I have also tried to make some deeper substantive changes to clarify some of the main points of the essay. Within the intro, I have attempted to define more clearly the parameters for analysis under a cosmological interpretive strategy. In addition, I agree that the essay may have felt too divided between the literary theories and the literary analysis. I was attempting to model how many theorists use literary analysis to demonstrate the tenets of their theories. Because of that divide, I attempted to revise and add in statements to connect the quotes from the theorists to their purpose within the analysis of the two plays. I will leave it up to the reviewers and editors to determine if those revisions are adequate.

                Finally, with regard to the reviewer’s comments about incorporating epic theater. While I agree that many of the techniques used such as minimalist staging and non-linear narratives are techniques used in epic theater, revising the essay to incorporate the development of epic theater and its relationship to the plays analyzed is out of the scope of this essay’s purpose. Writing an essay about epic theater would be the reviewer’s preferred essay, not mine. Also, epic theater is a genre of theater and not a literary theory. Epic theater’s focus is on technique and theme rather than ontological analysis. Also, Brechtian epic theater, in particular, appears to be informed and founded upon the same Marxist ideals as Frederic Jameson’s philosophies, and therefore the same critiques I make about Jameson in my essay, I would also make about Brechtian epic theater. Therefore, I respectfully decline to incorporate such a discussion of epic theater in my current essay. However, I do plan to compose more research and will definitely incorporate these discussions into the next iteration of critical analysis using this cosmological approach. I truly appreciate the feedback, it has made me a much better writer. Thank you!

Reviewer 2 Report

This is a soldi essya that will be helpful to readers of Wilder's and Miller's work. Specific errors pointed out  below should be corrected. Also, the author should see if sampling more recent theorists would enliven the argument a bit. 

“Dwight Eisenhauer” should be “Drew Eisenhauer.”

  Unnecessary comma in line 720. The opening sentence to the final paragraph, with a multitude of adjectives , does not work.   Line 21 should be ‘seen as both being deeply’   Line 56/57 should be ‘by which’ not ‘with which’   Line 684 and 685 The author should choose either “reader” or “audience” and not both.   Beautiful argument is perfectly reasonable, the a bit mechanistic at times. I wish the author did not need to use the multitude adjectives when one or two would do, it makes it seem a bit of an exercise. There is nalso the issue that I think people have always been open to spiritual interpretation of Thornton gWilder‘s drama, but Arthur Miller is seen as more of a naturalist or even a socialist realism playwright, so these differences in reception should be addressed. Also, although there is the newest collection by Amy Elias that is mentioned, much of the material the author draws on is from a somewhat dated attempt to do constructive postmodernism, whereas a newer studies object oriented ontology, speculative realism and radical theology would be just as pertinent.

Author Response

Thank you for your critique! I have made all of the grammatical and syntactic changes requested. I have also tried to make some deeper substantive changes to clarify some of the main points of the essay. Regarding the addition of more recent theorists, I agree that there has been much more published in the last few years that have built on the critiques many constructive postmodern theorists have discussed for years. However, just like how some people still prefer to conduct a Freudian reading of a text in spite of the innovations of Jung and Lacan, I preferred to focus on the constructive postmodern theorists since I view them as foundational to many of the more recent developments in literary theory such as cosmopolitanism, cosmodernism, and object-oriented ontology. I do plan to conduct further research and will make sure to include more recent publications and criticisms.

Round 2

Reviewer 1 Report

Revisions have contributed to more clarity of argument, particularly in the introduction. Seeing as the term "theopoetic" to describe the kind of cosmology the author outlines is central, particularly in the final analytical part and conclusion, the term should also appear in the introduction.

Author Response

I would like to thank the reviewer for the comments. They have been very helpful! I have added a definition of theopoetics to the introduction. I have also added to the introduction when appropriate, various statements referring back to the idea of theopoetics with regard to the discussion of Our Town and The Man Who Had All of the Luck. Please let me know if there are any other revisions needed. Thank you!
